# Mapping of a Major-Effect Quantitative Trait Locus for Seed Dormancy in Wheat

**DOI:** 10.3390/ijms25073681

**Published:** 2024-03-26

**Authors:** Yu Gao, Linyi Qiao, Chao Mei, Lina Nong, Qiqi Li, Xiaojun Zhang, Rui Li, Wei Gao, Fang Chen, Lifang Chang, Shuwei Zhang, Huijuan Guo, Tianling Cheng, Huiqin Wen, Zhijian Chang, Xin Li

**Affiliations:** College of Agronomy, Shanxi Key Laboratory of Crop Genetics and Molecular Improvement, Shanxi Agricultural University, Taiyuan 030031, China; gy980505@gmail.com (Y.G.);

**Keywords:** germination percentage, germination index, QTL mapping, RIL population, SNP array

## Abstract

The excavation and utilization of dormancy loci in breeding are effective endeavors for enhancing the resistance to pre-harvest sprouting (PHS) of wheat varieties. CH1539 is a wheat breeding line with high-level seed dormancy. To clarify the dormant loci carried by CH1539 and obtain linked molecular markers, in this study, a recombinant inbred line (RIL) population derived from the cross of weak dormant SY95-71 and strong dormant CH1539 was genotyped using the Wheat17K single-nucleotide polymorphism (SNP) array, and a high-density genetic map covering 21 chromosomes and consisting of 2437 SNP markers was constructed. Then, the germination percentage (GP) and germination index (GI) of the seeds from each RIL were estimated. Two QTLs for GP on chromosomes 5A and 6B, and four QTLs for GI on chromosomes 5A, 6B, 6D and 7A were identified. Among them, the QTL on chromosomes 6B controlling both GP and GI, temporarily named *QGp/Gi.sxau-6B*, is a major QTL for seed dormancy with the maximum phenotypic variance explained of 17.66~34.11%. One PCR-based diagnostic marker *Ger6B-3* for *QGp/Gi.sxau-6B* was developed, and the genetic effect of *QGp/Gi.sxau-6B* on the RIL population and a set of wheat germplasm comprising 97 accessions was successfully confirmed. *QGp/Gi.sxau-6B* located in the 28.7~30.9 Mbp physical position is different from all the known dormancy loci on chromosomes 6B, and within the interval, there are 30 high-confidence annotated genes. Our results revealed a novel QTL *QGp/Gi.sxau-6B* whose CH1539 allele had a strong and broad effect on seed dormancy, which will be useful in further PHS-resistant wheat breeding.

## 1. Introduction

Plant seeds can evade external adverse factors through dormancy, which is significant for their reproduction [1]. Common wheat (*Triticum aestivum* L., 2n = 6x = 42; AABBDD) is a widely cultivated crop worldwide and the first staple food crop domesticated in history, contributing about one-fifth of the total calories and proteins of the daily dietary intake of global population [2]. It originated from hybridization between cultivated tetraploid emmer wheat (*Triticum dicoccoides* L., 2n = 4x = 28; AABB) and *Aegilops tauschii* (2n = 2x = 14; DD) approximately 8000 years ago. The cultivation and domestication of common wheat have been directly associated with the spread of agriculture and settled societies [3]. In order to obtain the uniformity of seed germination and seedling emergence, the seed dormancy of wheat has gradually weakened in the process of domestication by human ancestors and extensive modern artificial selection, which results in pre-harvest sprouting (PHS) of physiologically mature seeds in spikes of most modern cultivars when the harvest time coincides with untimely rainfall, leading to losses in grain yield and deterioration in grain milling and baking quality [1,2,3]. Therefore, enhancing the level of seed dormancy in wheat to a certain extent is a necessary measure to address the negative impact on yield and quality caused by PHS.

Unearthing dormancy loci is fundamental endeavor in enhancing seed dormancy. Numerous genes associated with dormancy have been identified in wheat. The *Red-1* (*R-1*) gene controlling grain color was first observed to be positively associated with seed dormancy due to the fact that red grains usually have stronger dormancy characteristics than white grains [4]. *R-1* gene encodes the MYB transcription factor and has three sub-genomic copies, namely *Myb10-A*, *Myb10-B*, and *Myb10-D* [5]. Recent research revealed that *Myb10-D* confers PHS resistance by enhancing abscisic acid (ABA) biosynthesis to delay germination in wheat [6]. Overexpression of wheat *DELAY OF GERMINATION-1* (*TaDOG1*), the homolog of *AtDOG1* in Arabidopsis [7], significantly increases seed dormancy levels [8]. Wheat *MOTHER OF FT AND TFL 1* (*TaMFT*), the homolog of *AtMFT* [9], finely regulates seed dormancy by coordinating the ABA and gibberellic acid signaling pathways [10]. Moreover, wheat genes *MITOGEN-ACTIVATED PROTEIN KINASE KINASE 3* (*TaMKK3*) [11], *PLASMA MEMBRANE 19* (*TaPM19*) [12], and *SEED DORMANCY* (*TaSdr*) [13] were identified to play critical roles in the regulation of seed germination.

Moreover, transcriptome sequencing was applied in mining candidate genes for seed dormancy. Comparing the transcriptomes of seeds from two wheat varieties, strong dormant Scotty (PI 469294) and weak dormant Doublecrop (Cltr 17349), 2368 differentially expressed genes (DEGs) were identified at 20 days after anthesis, and several dormancy-related genes such as *TaMFT* and wheat *FLOWERING LOCUS C* (*TaFLC*) were selected and validated through qRT-PCR [14]. Additionally, a total of 13,154 DEGs were identified in four comparison groups between wheat cultivars Woori and Keumgang, with high and low levels of seed dormancy, respectively, and nine DEGs related to the spliceosome and proteasome pathways were verified as candidate genes using qRT-PCR [15]. A similar analysis on strong dormant AC Domain and weak dormant RL4452 showed that DEGs including wheat *LONELY GUY* (*TaLOG*), *cis ZEATIN-O-GLUCOSYLTRANSFERASE* (*TacZOG*), *ALDEHYDE OXIDASE* (*TaAO*), *UBIQUITIN* (*TaRUB*), and *AUXIN RESPONSE FACTOR* (*TaARF*) are involved in seed dormancy during the early stage of seed maturation [16]. Transcriptomic comparison of germination seeds and dormant seeds of the wheat variety MingXian169 revealed 3027 DEGs, and qRT-PCR confirmed that that the expressions of *ASCORBATE PEROXIDASE* (*APX*), *MONODEHYDROASCORBATE REDUCTASE* (*MDHAR*), *β*-*GLUCOSIDASE* (*GLU*), and *β*-*AMYLASE* (*AMY*) are significantly upregulated in germination seeds [17].

Furthermore, a multitude of dormancy- or PHS-associated quantitative trait loci (QTLs) have been identified on each chromosome of wheat [18,19]. Taking chromosome 6B as an example, molecular markers *wmc104* [20] and *wPt0959* [21] on the short arm of the chromosome were reported to be associated with tolerance to PHS in a recombinant inbred line (RIL) population and a set of 96 winter wheat accessions, respectively. In addition, *QPhs.spa-6B* [22] with 25% phenotypic variance explained (PVE) for germination index (GI) was identified in a doubled haploid mapping population; *QPhs.umb-6B* [23] with the PVE value of 3.09~4.33% for speed of germination index (SGI) was mapped in the doubled haploid population; and *Qphs.ahau-6B* was related to GI, field sprouting, and a period of dormancy in different environments with the *R*^2^ of 6.1~16.7% based on a genome-wide association study for 192 wheat varieties [24].

It is believed that the PHS resistance of wheat is predominantly due to dormancy, and the early interruption of seed dormancy has been considered the major component of PHS [13]. A lack of high dormancy level in many wheat cultivars during harvest period results in serious economic losses due to the adverse effects of pre-harvest sprouted wheat grains on end-product quality. Enhancing the tolerance to PHS is a major breeding objective in the world. Therefore, an understanding of the genetic control of seed dormancy and the development of functional markers are very necessary for marker-assisted breeding targeting for PHS tolerance in wheat breeding. CH1539 is a wheat breeding line with strong seed dormancy characteristics bred by our team. In order to clarify its dormant loci and obtain linked molecular markers, we created an RIL population derived from the intercrossing of CH1539 (CH) with the weakly dormant wheat line SY95-71 (SY). Here, this RIL population was used to construct a high-density genetic map using the Wheat17K SNP chip. Then, the germination percentage (GP) and GI of the seeds from each RIL were estimated, and the QTLs for GP and GI were mapped. We detected a major QTL for seed dormancy on chromosome 6B and developed its PCR-based diagnostic marker, which will be useful in further QTL-cloning and the PHS-resistant wheat breeding.

## 2. Results

### 2.1. Assessment of Seed Dormancy for SY and CH

The two breeding lines SY and CH have large differences in seed dormancy (Figure 1). SY had a weak seed dormancy with a GP of 52.42~70.21% and a GI of 28.30~34.40, whereas CH had a strong seed dormancy with a GP of 0~2.17% and a GI of 0~0.62. The best linear unbiased estimation (BLUE) values of both GP and GI in SY were significantly higher than those in CH (*p* < 0.001) (Table 1).

### 2.2. Phenotypic Variance of GP and GI in the SY × CH RILs

GP and GI showed segregations in the RIL population derived from the cross of SY and CH, which ranged from 0 to 100% and from 0 to 98.00, with the coefficient of variation values of 0.75~0.88 and 0.83~0.97, respectively (Table 1). According to the phenotypic data of three repeat tests and the derived BLUE datasets, GP and GI in the RILs exhibited continuous distribution (not normal distribution) and transgressive segregation, with some lines having stronger germination than SY, indicating that seed dormancy was a complex trait controlled by multi-genes (Figure 2).

### 2.3. Genetic Mapping of GP and GI

A total of 3328 SNPs from the 17K SNP chip showed allelic variation between SY and CH, and 2437 of which were mapped in the SY × CH RIL population after removing the redundant or excessive missing data. These SNP markers were assembled into 21 chromosomes to form a genetic map for the RILs with a marker density of 1.93 cM per marker. Based on the linkage map and the phenotypic BLUE data, two genomic regions on chromosomes 5A and 6B were found to have significant effects on the GP trait, with a PVE of 5.78% and 17.66%, respectively; and four genomic regions on chromosomes 5A, 6B, 6D and 7A were found to have significant effects on the GI trait, with a PVE of 6.41%, 34.11%, 5.06% and 9.02%, respectively (Table 2). The QTLs on chromosomes 5A and 6B controlled both GP and GI, and their additive effects came from weak dormant SY and strong dormant CH, respectively. Among them, the QTL on chromosomes 6B, temporarily named *QGp/Gi.sxau-6B*, has the maximum PVE values and negative additive effects, which indicates that *QGp/Gi.sxau-6B* is a major QTL for aggravating seed dormancy (Table 2).

### 2.4. Verification of QGp/Gi.sxau-6B

To verify the mapping results, PCR-based markers for *QGp/Gi.sxau-6B* were developed. *QGp/Gi.sxau-6B* was mapped to a 0.1 cM interval flanked by markers *995614* (chr.6B: 28,712,133) and *1091526* (chr.6B: 30,888,979), corresponding to a physical range of 2.18 Mbp (Figure 3a,b). We randomly designed 25 pairs of primers within the genome segment that ranged from 1 Mbp upstream of *995614* to 1 Mbp downstream of *1091526*, to amplify parental DNA and then sequence the products. Fortunately, sequence differences of insertion/deletion (InDel) were found on both sides and in the middle of *QGp/Gi.sxau-6B*. For the InDel2 on the outer site of SNP marker *995614*, there was a 14 bp deletion in SY but a 14 bp insertion in CH; for the InDel5 on the outer site of SNP marker *1091526*, there was a 10 bp deletion in SY but a 10 bp insertion in CH; and for the InDel3 located inside the *QGp/Gi.sxau-6B*, there was a 29 bp deletion in SY but a 29 bp insertion in CH (Figure 3b). Based on these InDels, three PCR markers *Ger6B-2*, *Ger6B-3* and *Ger6B-5* were developed and showed the expected parental polymorphisms by polyacrylamide gel electrophoresis (Figure 3c).

Verification results in the RIL population showed that the SY allele of *Ger6B-2*, *Ger6B-3* and *Ger6B-5* all corresponded to the higher GP and GI values, while the CH allele of the three markers corresponded to the lower values, confirming the effect of *QGp/Gi.sxau-6B*. The phenotypic differences between the two alleles of *Ger6B-3* (*p* < 0.0001) within the QTL were more significant than that of markers *Ger6B-2* and *Ger6B-5* (*p* < 0.01) on both sides (Figure 4). Hence, *Ger6B-3* was used as the diagnostic marker for allelic variation in *QGp/Gi.sxau-6B*.

### 2.5. Distribution of QGp/Gi.sxau-6B Alleles in Wheat

*Ger6B-3* was used to amplify a set of wheat germplasm containing 97 varieties. The results showed that 39 varieties carried the SY allele of *QGp/Gi.sxau-6B* and 58 varieties carried the CH allele of *QGp/Gi.sxau-6B*. The GP and GI values corresponding to the SY allele were significantly higher than those corresponding to the CH allele, with *p* < 0.01 and *p* < 0.05, respectively, indicating that *QGp/Gi.sxau-6B* had a wide effect on the dormancy of different wheat germplasms (Figure 5). Further analysis revealed that the distribution frequency of the strong-dormancy *QGp/Gi.sxau-6B* CH allele was high in landraces (78.95%), but decreased in cultivars (47.46%) (Figure 5c), suggesting that artificial selection in wheat breeding significantly affected seed dormancy ability and was one of the reasons for the decrease in resistance to PHS.

## 3. Discussion

### 3.1. CH1539 Can Be Used to Improve the Resistance of Cultivars to PHS

Modern wheat cultivars generally have poor resistance to PHS, as the breeding process artificially weakens the dormancy ability of their seeds to obtain quick and consistent seedling emergence after sowing. An improvement in seeds’ dormancy of different varieties can enhance their resistance to PHS. CH1539 is a strong dormant breeding line. In this study, the GP of CH1539 seeds was only 0~2.17% after 7 days of germination under suitable conditions of moisture and temperature, indicating that CH1539 will show effective resistance to PHS in the field when encountering continuous rainfall before harvest. CH1539 carries a major-dormancy QTL, *QGp/Gi.sxau-6B*, with a PVE of 17.66~34.11%. The distribution frequency of *QGp/Gi.sxau-6B* CH allele in cultivars is significantly lower than that in landraces, suggesting that the dormant effect of this locus has been attenuated during the breeding process and needs to be strengthened in the future. We developed the PCR-based diagnostic marker *Ger6B-3* for molecular-assisted selection of *QGp/Gi.sxau-6B*. In addition, CH1539 also carries the leaf rust resistance gene *LrCH1539* with the co-separated InDel-marker *sxau-2BS210* [25], which can be used for aggregate breeding of disease- and PHS resistance.

### 3.2. QGp/Gi.sxau-6B Is a Novel Dormancy Locus

*QGp/Gi.sxau-6B* was located in the chr.6B:28.7~30.9 Mbp physical position flanked by SNP markers *995614* and *1091526*. Currently, a total of five loci related to dormancy or PHS have been reported on chromosome 6B (Figure 6). Among them, *QPhs.spa-6B* was mapped in a *Xgwm508*-*Xgdm113* interval corresponding to the physical position range from chr.6B:47.8 Mbp to 77.1 Mbp [22], and it should be in the same locus as a PHS-associated single marker *wPt-0959* [21], but different from the position of *QGp/Gi.sxau-6B*. Moreover, the other PHS-linked single marker *Xwmc104* was located in the chr.6B:149.2 Mbp physical position [20], while *QPhs.umb-6B* [23] and *QPhs.ahau-6B* [24] were mapped to the physical locations chr.6B:439.4~567.6 Mbp and chr.6B:574.6~576.8 Mbp on the long arm of chromosome 6B, respectively. To sum up, *QGp/Gi.sxau-6B* is a novel locus that is different from all the dormancy loci reported on chromosomes 6B.

### 3.3. Prediction of Causing Gene for QGp/Gi.sxau-6B

There are thirty high-confidence genes in the *QGp/Gi.sxau-6B* interval according to the annotation of coding sequences (RefSeq v1.1) in the Chinese Spring genome database, five of which were highly expressed in germinated seeds but lowly expressed in dormant seeds based on published data [17]. Among these differentially expressed genes (DEGs), two genes *TraesCS6B02G049100* and *TraesCS6B02G049300* encoded histone H2B, two genes *TraesCS6B02G049400* and *TraesCS6B02G049500* encoded histone H2A, and the remain gene *TraesCS6B02G050700* encoded carboxypeptidase (CP) (Figure 6).

It has been established that histone H2A and H2B play important roles in the regulation of seed dormancy [26,27,28]. The epigenetic factor POWERDRESS (PWR) can interact with ABA-INSENSITIVE 3 (ABI3) and HISTONE DEACETYLASE 9 (HDA9) to reduce histone acetylation level and increase H2A deposition at the SOMNUS (SOM) locus, a positive regulator of seed dormancy, thus repressing the expression level of *SOM* and promoting seed germination process [29]. Moreover, histone H2B monoubiquitylation regulated by HISTONE MONOUBIQUITINATION1 (HUB1) and HUB2 can increase *DOG1* expression and then regulate the seed dormancy level [30,31,32]. Therefore, the *H2A* and *H2B* genes exhibit significant transcriptional changes during the epigenetic regulation of seed dormancy in wheat.

Additionally, the expression level of the *CP* gene *TraesCS6B02G050700* was significantly upregulated in germinated seeds. Carboxypeptidase is a key hydrolytic enzyme during germination in cereal grains [33,34]. In wheat, *CP* genes were involved in the acidification of endosperm starchy [35], mobilization of endosperm storage proteins [36], and other biochemical reactions during germination processes [37]. More importantly, *TraesCS6B02G050700* was highly expressed specifically in grains (Appendix A), indicating that it may be the causing gene for *QGp/Gi.sxau-6B*. Further experiments are needed to confirm this speculation.

## 4. Materials and Methods

### 4.1. Plant Materials

The wheat breeding line CH1539 with a high level of seed dormancy was developed by the College of Agronomy of Shanxi Agricultural University [25], and the other breeding line, SY95-71, with a low level of seed dormancy, was bred by the Triticeae Research Institute of Sichuan Agricultural University in the 1990s. A RIL population with 184 F_2:10_ lines derived from the cross of weak dormant SY95-71 and strong dormant CH1539 was used for mapping the dormancy sites.

A set of wheat germplasm comprising 97 hexaploid accessions (*Triticum aestivum* L.) [38], including 38 landraces and 59 modern cultivars (Appendix A), was used to estimate the adaptability and the allele distribution of QTL. All the plant materials were planted in the greenhouse at Shanxi Agricultural University (Taiyuan, China) in 2022.

### 4.2. Seed Dormancy Test

The seed germination of each RIL and of each wheat germplasm was evaluated following the method as described before [22,23]. Briefly, spikes at physiological maturity characterized as a loss of green color from the glumes of the basal spikelets were harvested, naturally air-dried for 5 days, hand threshed to avoid damaging the embryos and seed coat, and then temporarily stored at –20 °C for about 10 days to maintain dormancy. After the harvesting was complete, all seeds were taken out from the freezer to room temperature and air-dried again for 2 days. Germination tests were performed on paper filters moistened with distilled water placed in sterile plastic Petri dishes at 20 °C in the dark for 7 days. A seed was considered germinated when the coleorhiza emerged beyond the seed coat. Germinated seeds were counted daily and removed, and then the GP and GI of each line were calculated from the following formula, respectively:GP (%) = (*n*_1_ + *n*_2_ + *n*_3_ + ⋯ + *n*_7_)/*N* × 100%;
GI = (7 × *n*_1_ + 6 × *n*_2_ + 5 × *n*_3_ + ⋯ + 1 × *n*_7_) × 100/(7 × *N*)
where *N* is the number of total seeds, which was 50 in this study, and *n*_1_, *n*_2_, *n*_3_, …, *n*_7_ are the number of germinated seeds on the first, second, third, and subsequent days until the seventh day, respectively. The test was independently repeated three times, and then the BLUE values were calculated by Genstat v22 software (HZRC Technology Co., Ltd., Beijing, China).

### 4.3. Genotyping and QTL Mapping

The genomic DNA of each RIL and the parents was extracted from leaf samples at seedling stage and then genotyped using a Wheat17K SNP chip with 17,526 markers at Diversity Arrays Technology Pty Ltd. (Canberra, Australia). Genotypic data were analyzed as described previously [39]. Briefly, the SNP markers exhibiting a base difference between SY95-71 and CH1539 were screened. Then, these parent-polymorphic markers with physical positions identified on the Chinese Spring genome (International Wheat Genome Consortium RefSeq v1.0), had less than 20% missing values, and contained different recombinant information were selected to construct linkage groups using the MAP program of IciMapping v4.0 software (Chinese Academy of Agricultural Sciences, Beijing, China). Then, the genetic map was integrated with GP- and GI-BLUE values of RILs for QTL analysis using the Kosambi function under the BIP program of ICIMapping v4.0 software, setting the threshold of logarithm of the odds as 2.5.

### 4.4. Diagnostic Marker Developing

PCR-based diagnostic markers for QTL were developed as described previously [40]. Firstly, primer pairs were randomly designed within the genome section extending 1 Mbps on each side of the target QTL and then amplified genomic DNA of SY and CH. These amplified products were sequenced to find sequence differences. Afterwards, new InDel-markers *Ger6B-2* (F: 5′-CACACAAGCGGTGCCTGC-3′; R: 5′-GTACACGCATGTGATGATGTATC-3′), *Ger6B-3* (F: 5′-CCCTAGCATCGAGAGTTCT-3′; R: 5′-ACATCACCACGTTATGCCG-3′), and *Ger6B-5* (F: 5′-ATCACGCGCATGTCGCAC-3′; R: 5′-GTGTATGAGAAGATAAAGGGAAG-3′) were developed based on the parental sequence differences, and their amplification conditions were as follows: 94 °C for 1 min, 38 cycles of 94 °C for 30 s, 62 °C for 30 s, and then 72 °C for 30 s, with a final extension of 72 °C for 5 min. The PCR products were differentiated by polyacrylamide gel electrophoresis at 180 V for 1.5 h. The InDel-marker with the most significant phenotypic differences between the two alleles was used as the diagnostic marker for QTL allelic variation.

### 4.5. Analysis of QTL Alleles in Germplasms

GP and GI were tested for the 97 wheat accessions according to the method described in Section 4.2. Then, allelic variation in target QTL in germplasm was identified using the diagnostic marker to evaluate the correlation between seed dormancy and the target QTL alleles. The distribution frequency of each allele in landraces and cultivars was also counted. In addition, the fragments per kilobase of exon model per million mapped fragments (FPKM) values of high-confidence annotated genes within target QTL in germinated seeds (GS) and dormant seeds (DS) of wheat variety Mingxian169 were downloaded from published data [17], and then DEGs between GS and DS were identified on the BMKCloud platform (www.biocloud.net (accessed on 18 January 2024)) with a threshold of |log2FC| ≥ 1 and FDR ≤ 0.01 (Appendix A) and selected for visualization on heat map.

### 4.6. Statistical Analysis

The Origin v3.1 software (OriginLab, Northampton, MA, USA) was used to perform the statistical analysis by one-way analysis of variance (ANOVA), and *p* < 0.05 was considered a statistically significant difference, while *p* < 0.01 was considered an extremely statistically significant difference.

## 5. Conclusions

This study clarified the dormant loci carried by CH1539 and obtained close-linked molecular markers. Two QTLs for GP on chromosomes 5A and 6B and four QTLs for GI on chromosomes 5A, 6B, 6D and 7A were identified. Among them, *QGp/Gi.sxau-6B* controlling both GP and GI was a major QTL for seed dormancy with the maximum PVE of 17.66~34.11%. One PCR-based diagnostic marker *Ger6B-3* for *QGp/Gi.sxau-6B* was developed, and the genetic effect of *QGp/Gi.sxau-6B* on the RIL population and wheat germplasm was successfully confirmed.

## Figures and Tables

**Figure 1 ijms-25-03681-f001:**
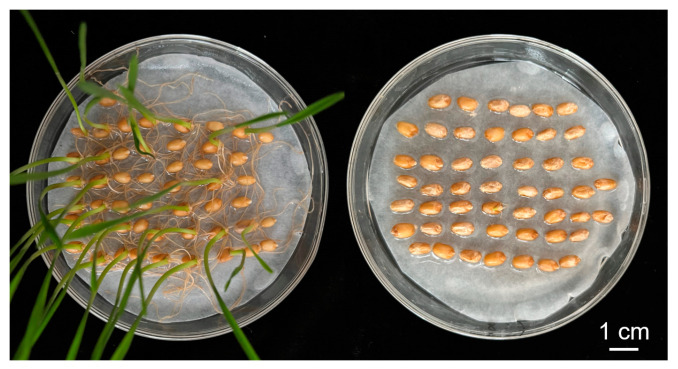
Wheat breeding lines SY95-71 (**left**) and CH1539 (**right**) after 7 days of germination.

**Figure 2 ijms-25-03681-f002:**
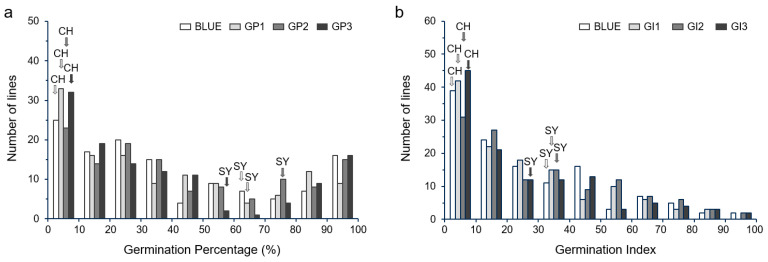
Variation in seed dormancy in SY × CH RIL population in three repeat tests and the BLUE dataset. (**a**) Germination percentage (GP) of RILs. (**b**) Germination index (GI) of RILs. CH: CH1539; SY: SY95-71.

**Figure 3 ijms-25-03681-f003:**
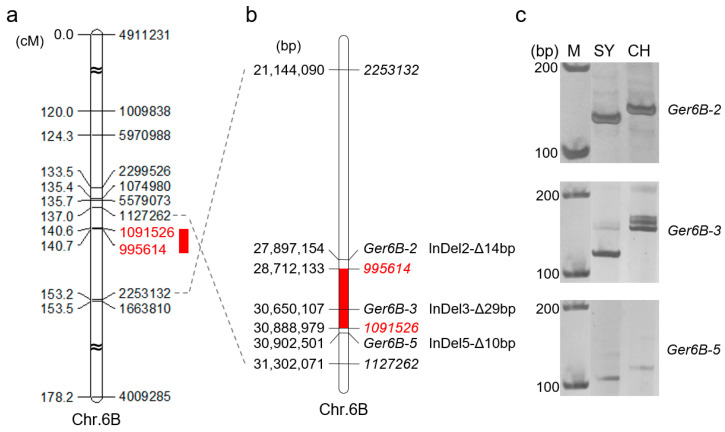
Map position and linked markers of *QGp/Gi.sxau-6B*. (**a**) Genetic map. (**b**) Genomic map. (**c**) Results of polyacrylamide gel electrophoresis for PCR markers. Red boxes represent *QGp/Gi.sxau-6B*, and the linked SNP markers are marked in red. M: Marker ladder; CH: CH1539; SY: SY95-71.

**Figure 4 ijms-25-03681-f004:**
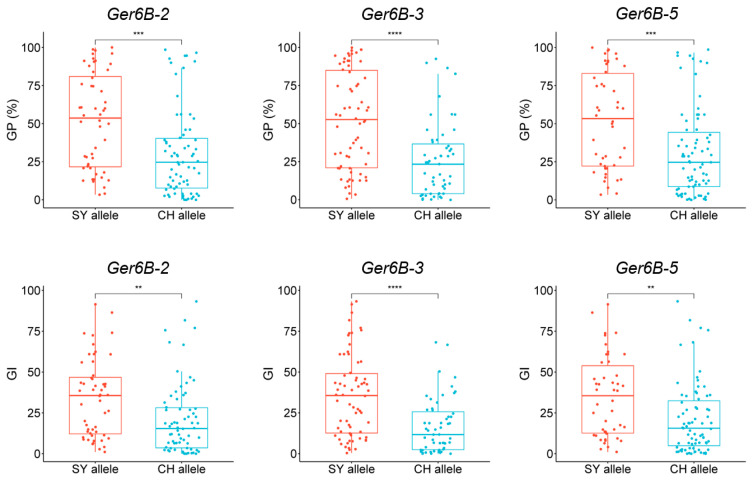
Significant differences analysis in genotypes of the three PCR-markers of *QGp/Gi.sxau-6B* in SY × CH RIL population. CH: CH1539; SY: SY95-71; GP: germination percentage; GI: germination index. Red and blue represent SY allele and CH allele of marker. ** indicates *p* < 0.01, *** indicates *p* < 0.001, and **** indicates *p* < 0.0001.

**Figure 5 ijms-25-03681-f005:**
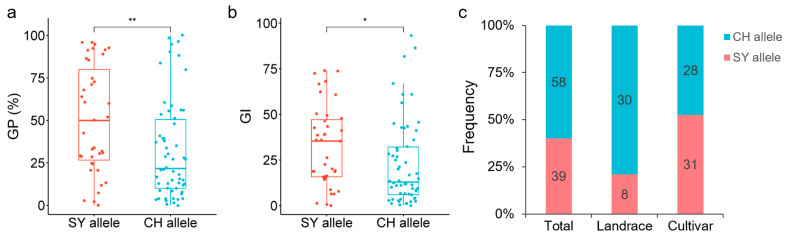
Effect and distribution frequency of *QGp/Gi.sxau-6B* alleles in wheat germplasm. (**a**,**b**) Germination percentage (GP) and germination index (GI) of the two alleles of *QGp/Gi.sxau-6B* in 97 wheat varieties. (**c**) Distribution frequency of the two alleles of *QGp/Gi.sxau-6B* in 97 varieties, including 38 landraces and 59 cultivars. Red and blue represent SY allele and CH allele of marker *Ger6B-3*. * indicates *p* < 0.05 and ** indicates *p* < 0.01.

**Figure 6 ijms-25-03681-f006:**
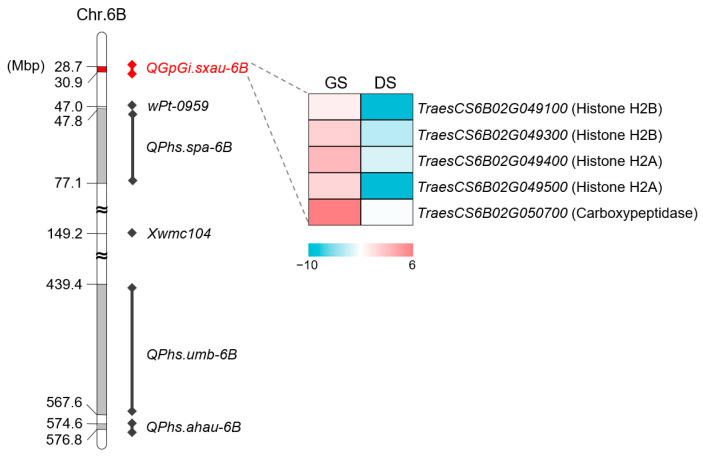
The physical position of *QGp/Gi.sxau-6B* compared with the previously reported dormancy- or PHS-associated loci on chromosome 6B. The boxes or bars indicate intervals harboring QTLs flanked by markers, and the rhombic dots indicate single markers. Differentially expressed genes between germinated seeds (GS) and dormant seeds (DS) in *QGp/Gi.sxau-6B* region are selected from published data [17] and visualized by heatmaps.

**Table 1 ijms-25-03681-t001:** Assessment of seed dormancy for SY × CH RIL population and the parents.

Trait	Repeat	Parents		RIL Population			
		SY	CH	Min	Max	Mean	CV
GP (%)	1	66.54	0	0	100.00	38.11	0.82
	2	70.21	2.17	0	100.00	43.06	0.75
	3	52.42	0	0	100.00	38.22	0.88
	BLUE	63.06	0.72 ***	0	100.00	39.80	0.82
GI	1	33.74	0	0	89.14	25.12	0.91
	2	34.40	0.62	0	95.14	30.06	0.83
	3	28.30	0	0	98.00	25.17	0.97
	BLUE	32.15	0.21 ****	0	94.10	26.78	0.90

CH: CH1539; SY: SY95-71; GP: germination percentage; GI: germination index; CV: coefficient of variation; BLUE: best linear unbiased estimation; ***: *p* < 0.001; ****: *p* < 0.0001.

**Table 2 ijms-25-03681-t002:** QTL mapping for GP and GI in SY × CH RIL population.

Trait	Chromosome	Position (cM)	Left Marker	Right Marker	LOD	PVE (%)	ADD
GP	5A	138	*996462*	*3064715*	2.71	5.78	4.98
	6B	141	*995614*	*1091526*	6.30	17.66	–13.78
GI	5A	140	*3064715*	*1207181*	3.32	6.41	2.82
	6B	141	*995614*	*1091526*	11.84	34.11	–17.14
	6D	87	*1239681*	*12736348*	2.83	5.06	–2.48
	7A	17	*3064775*	*1230235*	5.66	9.02	11.15

CH: CH1539; SY: SY95-71; GP: germination percentage; GI: germination index; LOD: logarithm of the odds; PVE: phenotypic variance explained; ADD: additive effect.

## Data Availability

Data are contained within this article and Appendix A.

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
