# Peer review of "Mapping of a Major-Effect Quantitative Trait Locus for Seed Dormancy in Wheat"

_ijms, 2024, doi:10.3390/ijms25073681_

Round 1
Reviewer 1 Report
Comments and Suggestions for Authors
Dear Authors,
Thank you very much for your study. I reviewed your manuscript, and ı suggested some corrections and additions to important topics. After the revision, I think that it will be better compared to the first.
The suggestions:
1. The "4.1. Plant Materials" section can be rewritten again to give detailed informations about the past of the wheat lines. Also this section also a little bit complex, it can be written more clearly.
2. Why did the authors did not give some references especially in sections "4.2. Seed Dormancy Test" and "4.4. Diagnostic Marker Developing".
3. The "4.3. Genotyping and QTL mapping" section can be rewritten as more detailed.
4. I think that the statistical analysis section is not enough. The analysis method can be rewritten by giving some information for the study.
The result section was written well, however some important information should be added. Also, the authors should add some new findings to support the results in the discussion part. Other needed corrections sent with previous evaluation as following;
1. The aim of the study at the end of the introduction part should be rewritten why the study was carried out?
2. The authors should give some information about wheat and its agriculture in the introduction part.
3. The references before the 2000 years can be updated.
4. The reference section should be controlled again for the "1. [1]", "2. [2]" and etc.
The language of the manuscript should be rechecked.
Author Response
- The "4.1. Plant Materials" section can be rewritten again to give detailed informations about the past of the wheat lines. Also this section also a little bit complex, it can be written more clearly.
Reply: Thanks for the suggestion. We rewritten the "4.1. Plant Materials" section as follow:
Wheat breeding line CH1539 with high level of seed dormancy was developed by the College of Agronomy of Shanxi Agricultural University, and the other breeding line SY95-71 with low level of seed dormancy was bred by the Triticeae Research Institute of Sichuan Agricultural University in the 1990s. A RIL population with 184 F2:10 lines derived from the cross of weak dormant SY95-71 and strong dormant CH1539 was used for mapping the dormancy sites. A set of wheat germplasm comprising 97 hexaploid accessions (Triticum aestivum L.), including 38 landraces and 59 modern cultivars (Table S1), was used to estimate the adaptability and the allele distribution of QTL. All the plant materials were planted in the greenhouse at Shanxi Agricultural University (Taiyuan, China) in 2022.
- Why did the authors did not give some references especially in sections "4.2. Seed Dormancy Test" and "4.4. Diagnostic Marker Developing".
Reply: References [22] and [23] was added for 4.2 section, and reference [40] was added for 4.4 section in the revision. - The "4.3. Genotyping and QTL mapping" section can be rewritten as more detailed.
Reply: Some details were added in 4.3 section: "Genomic DNA of each RIL and the parents was extracted from leaf samples at seedling stage and then genotyped using Wheat17K SNP chip with 17,526 markers at Diversity Arrays Technology Pty Ltd. (Canberra, Australia). Genotypic data was analyzed as described previously [39]. Briefly, the SNP markers exhibiting base difference between SY95-71 and CH1539 were screened. Then, these parents-polymorphic markers that with physical positions on the Chinese Spring genome (International Wheat Genome Consortium RefSeq v1.0) and had less than 20% missing values as well as contained different recombinant information were selected to construct linkage groups using the MAP program of IciMapping v4.0 software (Chinese Academy of Agricultural Sciences, Beijing, China)". - I think that the statistical analysis section is not enough. The analysis method can be rewritten by giving some information for the study.
Reply: The 4.6 section was rewritten as follow: " The Origin v3.1 software (OriginLab, Northampton, MA, USA) was used to perform the statistical analysis by one-way analysis of variance (ANOVA), and p < 0.05 was con-sidered a statistically significant difference, while p < 0.01 was considered an extremely statistically significant difference". - The result section was written well, however some important information should be added. Also, the authors should add some new findings to support the results in the discussion part. Other needed corrections sent with previous evaluation as following;
(1) The aim of the study at the end of the introduction part should be rewritten why the study was carried out?
Reply: We added the sentence “In order to clarify its dormant loci and obtain linked molecular markers, we created a RIL population derived from the intercrossing of CH1539 (CH) with the weakly dormant wheat line SY95-71 (SY)”.
(2) The authors should give some information about wheat and its agriculture in the introduction part.
Reply: We rewritten a sentence as follow: “Wheat (Triticum aestivum L., 2n = 6x = 42; AABBDD) is a widely cultivated crop worldwide and the first staple food crop domesticated in history, contributing about one-fifth of the total calories and proteins of daily dietary intake for global population”.
(3) The references before the 2000 years can be updated.
Reply: We updated the Ref "Washio, K.; Ishikawa, K. Organ-specific and Hormone-dependent Expression of Genes for Serine Carboxypeptidases During Development and Following Germination of Rice Grains. Plant Physiol. 1994, 105, 1275–1280." as" Li, Z.; Tang, L.; Qiu, J.; Zhang, W.; Wang, Y.; Tong, X.; Wei, X.; Hou, Y.; Zhang, J. Serine Carboxypeptidase 46 Regulates Grain Filling and Seed Germination in Rice (Oryza sativa L.). PLoS One 2016, 11, e0159737." The remaining three Refs ([20],[34],[35]) published before 2000 cannot be replaced. We hope to receive your understanding.
(4) The reference section should be controlled again for the "1. [1]", "2. [2]" and etc.
Reply: We made corresponding corrections.
(5) The language of the manuscript should be rechecked.
Reply: We rechecked the grammar and spelling of the manuscript.
Reviewer 2 Report
Comments and Suggestions for Authors
The manuscript titled: "Mapping of a Major-effect Quantitative Trait Locus for Seed Dormancy in Wheat" by Yu Gao Yu Gao, Linyi Qiao, Chao Mei, Lina Nong, Qiqi Li, Xiaojun Zhang, Rui Li, Wei Gao, Fang Chen, Lifang Chang, Shuwei Zhang, Huijuan Guo, Tianling Cheng, Huiqin Wen, Zhijian Chang and Xin Li provides a detailed exploration of dormancy loci in wheat breeding, with a specific focus on enhancing resistance to pre-harvest sprouting (PHS). The study utilizes a recombinant inbred line (RIL) population generated from the cross of two breeding lines, 'SY95-71' with weak dormancy and 'CH1539' with strong dormancy.
The researchers employed the Wheat17K single-nucleotide polymorphism (SNP) array to genotype the RIL population, resulting in a high-density genetic map covering all 21 chromosomes and comprising 2437 SNP markers. Subsequently, the germination percentage (GP) and germination index (GI) of seeds from each RIL were assessed, leading to the identification of significant quantitative trait loci (QTLs) for GP and GI on chromosomes 5A, 6B, 6D, and 7A.
Of particular interest is the major QTL located on chromosome 6B, designated as QGp/Gi.sxau-6B, which not only controls both GP and GI but also exhibits a substantial phenotypic variance explained ranging from 17.66% to 34.11%. The manuscript provides an in-depth analysis of this novel QTL, emphasizing its potential utility in enhancing seed dormancy and, consequently, contributing to pre-harvest sprouting resistance in wheat.
The researchers also developed a PCR-based diagnostic marker, Ger6B-3, specific to QGp/Gi.sxau-6B. This marker proved effective in confirming the genetic effect of QGp/Gi.sxau-6B in both the RIL population and a broader set of wheat germplasm, comprising 97 accessions. Notably, QGp/Gi.sxau-6B's physical position (28.7~30.9 Mbp) differs from known dormancy loci on chromosome 6B, highlighting its novelty in the context of wheat seed dormancy.
Nevertheless, I have a few minor comments that the authors should consider during the subsequent stages of manuscript preparation for publication:
1) Abstract - please briefly mention the research hypothesis.
2) Keywords - please list them in alphabetical order and do not include words appearing in the title of the manuscript.
3) Introduction - in my opinion, it provides a poor research background for such an important and popular problem all over the world - dormancy of wheat seeds. Lines 64-71 - in addition to the purpose of the research, a precisely formulated research hypothesis should be included here.
4) Results - all abbreviations used in tables and figures should be explained in the heading, below the table or in the legend, respectively; Figure 2 - please explain the meaning of the abbreviations CH and SY; Figure 3 - please explain the meaning of the abbreviations M, SY and CH; Figure 4 - please explain the meaning of the abbreviations GP and GI.
5) Materials and Methods - Line 224 - the scientific name of the tested species should also be used here. Lines 229-236 - please explain how long the seeds were stored at a temperature of -20 degree C. In how many repetitions was the germination test performed and how many seeds were tested in one repetition. What developmental phase were the seedlings that were removed from the petri dishes?
6) Conclusion - it is necessary to reword it - it is necessary to respond to the research hypothesis set at the beginning. Provide the practical application of the obtained results and determine the further direction of research.
7) Please adapt the entire manuscript better to the requirements of the template applicable in IJMS, and pay special attention to the References chapter.
In conclusion, the manuscript presents valuable insights into the genetic basis of seed dormancy in wheat, particularly with the identification and characterization of the major QTL QGp/Gi.sxau-6B. The development of a diagnostic marker adds practical significance to these findings, opening avenues for further research and application in pre-harvest sprouting-resistant wheat breeding programs. The comprehensive nature of the study and the novelty of the identified QTL make this manuscript a valuable contribution to the field of wheat genetics and breeding.
I believe the results presented in this manuscript are highly valuable, and the Editorial Team of IJMS should consider publishing them.
Author Response
- Abstract - please briefly mention the research hypothesis.
Reply: Thanks for the suggestion. We added one sentence “CH1539 is a wheat breeding line with high level seed dormancy. To clarify the dormant loci carried by CH1539 and obtain linked molecular markers…” in Abstract.
- Keywords - please list them in alphabetical order and do not include words appearing in the title of the manuscript.
Reply: We made corresponding corrections as follow: “germination percentage; germination index; QTL mapping; RIL population; SNP array”.
- Introduction - in my opinion, it provides a poor research background for such an important and popular problem all over the world - dormancy of wheat seeds.
Reply: We added a paragraph about research background of mining candidate genes for seed dormancy: “Moreover, transcriptome sequencing was applied in mining candidate genes for seed dormancy. Comparing the transcriptomes of seeds from two wheat varieties, strong dormant Scotty (PI 469294) and weak dormant Doublecrop (Cltr 17349), 2368 differential-ly expressed genes (DEGs) were identified at 20 days after anthesis, and several dorman-cy-related genes such as TaMFT and wheat FLOWERING LOCUS C (TaFLC) were selected and validated through qRT-PCR [14]. Additionally, a total of 13,154 DEGs were identified in four comparison groups between wheat cultivars Woori and Keumgang, with high and low level of seed dormancy, respectively, and nine DEGs related to the spliceosome and proteasome pathways were verified as candidate genes using qRT-PCR [15]. Similar analysis on strong dormant AC Domain and weak dormant RL4452 showed that DEGs including wheat LONELY GUY (TaLOG), cis ZEATIN-O-GLUCOSYLTRANSFERASE (TacZOG), ALDEHYDE OXIDASE (TaAO), UBIQUITIN (TaRUB), and AUXIN RESPONSE FACTOR (TaARF) were involved in seed dormancy during the early stage of seed matura-tion [16]. Transcriptomic comparison of germination seeds and dormant seeds of wheat variety MingXian169 revealed 3027 DEGs, and qRT-PCR confirmed that the expressions of ASCORBATE PEROXIDASE (APX), MONODEHYDROASCORBATE REDUCTASE (MDHAR), β-GLUCOSIDASE (GLU), and β-AMYLASE (AMY) were significantly upregu-lated in germination seeds [17].”
- Lines 64-71 - in addition to the purpose of the research, a precisely formulated research hypothesis should be included here.
Reply: We made modification as: “In order to clarify its dormant loci and obtain linked molecular markers…”.
- Results - all abbreviations used in tables and figures should be explained in the heading, below the table or in the legend, respectively; Figure 2 - please explain the meaning of the abbreviations CH and SY; Figure 3 - please explain the meaning of the abbreviations M, SY and CH; Figure 4 - please explain the meaning of the abbreviations GP and GI.
Reply: We made corrections as suggestion.
- Materials and Methods - Line 224 - the scientific name of the tested species should also be used here.
Reply: “Triticum aestivum L.” was added.
- Lines 229-236 - please explain how long the seeds were stored at a temperature of -20 degree C.
Reply: We modified it as “and then temporarily stored at –20°C for about 10 days to maintain dormancy”.
- In how many repetitions was the germination test performed and how many seeds were tested in one repetition.
Reply: We modified it as “where N is the number of total seeds, which was 50 in this study” and “The test was independently repeated three times”.
- What developmental phase were the seedlings that were removed from the petri dishes?
Reply: We added a sentence “A seed was considered germinated when the coleorhiza emerged beyond the seed coat”.
- Conclusion - it is necessary to reword it - it is necessary to respond to the research hypothesis set at the beginning. Provide the practical application of the obtained results and determine the further direction of research.
Reply: We rewritten it as follow: “This study clarified the dormant loci carried by CH1539 and obtained the close linked molecular markers. Two QTLs for GP on chromosomes 5A and 6B and four QTLs for GI on chromosomes 5A, 6B, 6D and 7A were identified. Among them, QGp/Gi.sxau-6B controlling both GP and GI was a major QTL for seed dormancy with the maximum PVE of 17.66~34.11%. One PCR-based diagnostic marker Ger6B-3 for QGp/Gi.sxau-6B was de-veloped, and the genetic effect of QGp/Gi.sxau-6B in the RIL population and wheat germplasm was successfully confirmed”.
- Please adapt the entire manuscript better to the requirements of the template applicable in IJMS, and pay special attention to the References chapter.
Reply: We used the IJMS template for the revision as suggestion.
Round 2
Reviewer 1 Report
Comments and Suggestions for Authors
Dear Authors,
Thank you very much for your revised manuscript. I reviewed your manuscript again and saw some lack information about my suggestions. Please recheck them.
The lack suggestions are follow:
1. The result section was written well, however some important information should be added. Also, the authors should add some new findings to support the results in the discussion part. The reply for Q1 was not answered.
2. The aim of the study at the end of the introduction part should be rewritten why the study was carried out?
Thank you very much for your reply but the given answer is not enough. So, please write again.
3- The authors should give some information about wheat and its agriculture in the introduction part.
Reply: We rewritten a sentence as follow: “Wheat (Triticum aestivum L., 2n = 6x = 42; AABBDD) is a widely cultivated crop worldwide and the first staple food crop domesticated in history, contributing about one-fifth of the total calories and proteins of daily dietary intake for global population”.
The reply for the Q3 is not enough. Please add some specific detailes on wheat.
Author Response
- The result section was written well, however some important information should be added. Also, the authors should add some new findings to support the results in the discussion part. The reply for Q1 was not answered.
Reply: Thank you for the suggestion. Apologize for that we do not quite understand these two issues. For the RESULT section, we would like to know what specific information we need to provide. For the DISCUSSION section, does the “add some new findings” means we need to supplement new experiments?
We will immediately add all the relevant information once we receive your specific suggestions. However, we may have no time to perform new experiments as the first author needs this paper to apply for his master's degree on April. We sincerely hope you can understand.
- The aim of the study at the end of the introduction part should be rewritten why the study was carried out? Thank you very much for your reply but the given answer is not enough. So, please write again.
Reply: We rewritten this part as: “It is believed that PHS resistance of wheat is predominantly due to dormancy, and the early interruption of seed dormancy has been considered as the major component of PHS [13]. Lack of high dormant level in many wheat cultivars during harvest period resulted in serious economic losses due to the adverse effects of pre-harvest sprouted wheat grains on end-product quality. Enhancing of tolerance to PHS is a major breeding objective in the world. Therefore, an understanding of genetic control of seed dormancy and development of functional markers are very necessary for marker-assisted breeding targeting for PHS tolerance in wheat breeding. CH1539 is a wheat breeding line with strong seed dormancy characteristics bred by our team. In order to clarify its dormant loci and obtain linked molecular markers, we created a RIL population derived from the intercrossing of CH1539 (CH) with the weakly dormant wheat line SY95-71 (SY). Here, this RIL population was used to …”.
- The authors should give some information about wheat and its agriculture in the introduction part. The reply for the Q3 is not enough. Please add some specific detailes on wheat.
Reply: We rewritten this part as: “Common wheat (Triticum aestivum L., 2n = 6x = 42; AABBDD) is a widely cultivated crop worldwide and the first staple food crop domesticated in history, contributing about one-fifth of the total calories and proteins of daily dietary intake for global population [2]. It originated from hybridization between cultivated tetraploid emmer wheat (Triticum dicoccoides L., 2n = 4x = 28; AABB) and Aegilops tauschii (2n = 2x = 14; DD) approximately 8000 years ago. Cultivation and domestication of common wheat has been directly associated with the spread of agriculture and settled societies [3]”. And a new Ref [3] (Brenchley, R.; Spannagl, M.; Pfeifer, M.; Barker, G.L.; D'Amore, R.; Allen, A.M.; McKenzie, N.; Kramer, M.; Kerhornou, A.; Bolser, D.; et al. Analysis of the Bread Wheat Genome Using Whole-genome Shotgun Sequencing. Nature 2012, 491, 705–710) was replaced the old one (Soper, J.F.; Cantrell, R.G.; Dick, J.W. Sprouting Damage and Kernel Color Relationships in Durum Wheat. Crop Sci. 1989, 29, 895–898).
We really appreciate for all the suggestions. We think the main text of the manuscript has more than 4000 words now.
Round 3
Reviewer 1 Report
Comments and Suggestions for Authors
Dear Authors,
Thank you very much for your response to my suggestions. I accepted your response and good luck to first author needs this paper to apply for his master's degree on April.